# Statistical Complexity Analysis of Turing Machine tapes with Fixed Algorithmic Complexity Using the Best-Order Markov Model

**DOI:** 10.3390/e22010105

**Published:** 2020-01-16

**Authors:** Jorge M. Silva, Eduardo Pinho, Sérgio Matos, Diogo Pratas

**Affiliations:** 1Institute of Electronics and Informatics Engineering of Aveiro, University of Aveiro, 3810-193 Aveiro, Portugal; eduardopinho@ua.pt (E.P.); aleixomatos@ua.pt (S.M.); pratas@ua.pt (D.P.); 2Department of Electronics, Telecommunications and Informatics, University of Aveiro, 3810-193 Aveiro, Portugal; 3Department of Virology, University of Helsinki, 00100 Helsinki, Finland

**Keywords:** turing machines, information theory, statistical complexity, algorithmic complexity, computational complexity, Markov models, compression-based analysis

## Abstract

Sources that generate symbolic sequences with algorithmic nature may differ in statistical complexity because they create structures that follow algorithmic schemes, rather than generating symbols from a probabilistic function assuming independence. In the case of Turing machines, this means that machines with the same algorithmic complexity can create tapes with different statistical complexity. In this paper, we use a compression-based approach to measure global and local statistical complexity of specific Turing machine tapes with the same number of states and alphabet. Both measures are estimated using the best-order Markov model. For the global measure, we use the Normalized Compression (NC), while, for the local measures, we define and use normal and dynamic complexity profiles to quantify and localize lower and higher regions of statistical complexity. We assessed the validity of our methodology on synthetic and real genomic data showing that it is tolerant to increasing rates of editions and block permutations. Regarding the analysis of the tapes, we localize patterns of higher statistical complexity in two regions, for a different number of machine states. We show that these patterns are generated by a decrease of the tape’s amplitude, given the setting of small rule cycles. Additionally, we performed a comparison with a measure that uses both algorithmic and statistical approaches (BDM) for analysis of the tapes. Naturally, BDM is efficient given the algorithmic nature of the tapes. However, for a higher number of states, BDM is progressively approximated by our methodology. Finally, we provide a simple algorithm to increase the statistical complexity of a Turing machine tape while retaining the same algorithmic complexity. We supply a publicly available implementation of the algorithm in C++ language under the GPLv3 license. All results can be reproduced in full with scripts provided at the repository.

## 1. Introduction

Information can be thought of as some message stored or transmitted using some medium. The need to communicate efficiently led to the creation of written systems, which in contrast to other modes of communication, such as a painting, make use of a finite and discrete set of symbols to express a concept [1,2]. Any written language can be thought of in this way since messages are formed by combining and arranging symbols in specific patterns. The improvement of communication propelled the development of computing machines and consequentially, the science of information. The first well documented mechanical computer was the difference engine by Charles Babbage in 1822, which led to the development of the analytical engine around 1837, encapsulating most of the elements of modern computers [3,4]. A century later, a universal calculating machine (Turing machine) was introduced by Alan Turing [5].

### 1.1. Turing Machines

Turing’s proposal on automatic machines (Turing machines) simplified the concept of decision machines by defining an abstract machine that handles symbols on a strip of tape according to a table of rules. Despite the simplicity of the mathematical model, a Turing machine (TM) is capable of simulating the logic behind any computer algorithm provided to it [6]. Specifically, a TM is composed of a rule table, a head, and a state, and operates on a memory tape divided into discrete cells. The machine starts with an initial state and its head positioned at a given point of the tape. In this position, it scans a single symbol of the tape and based on the scanned symbol, initial state, and instruction table, the TM writes a symbol on the tape and changes its position. This process repeats continuously until the machine enters its final state, causing a halt in the computation [7].

Formally, a TM can be defined as a 7-tuple $T=\langle Q,\Gamma ,b,\Sigma ,\delta ,q_{0},F\rangle$, where: *Q* is a finite, non-empty set of *states*; Γ is a finite, non-empty set of *tape alphabet symbols*; b∈Γ is the *blank symbol*; Σ⊆Γ∖{b} is the set of *input symbols*; $q_{0}\in Q$ is the *initial state*; and F⊆Q is the set of *final states* or *accepting states*. Finally, the *transition function, δ* is characterized by δ:(Q∖F)×Γ¬→Q×Γ×{L,R,N}, where *L* is left shift, *R* is right shift, and *N* is no shift. The initial tape contents are said to be *accepted* by *T* if it eventually enters a *final state* of *F* and halts. Despite this definition, there are many variants of these machines. For instance, there are models which allow symbol erasure or no writing, models that do not allow motion [8], and others which discard the stop criteria or final state, and, consequentially, do not halt.

With this model, a mathematical description of a simple device capable of arbitrary computations was developed, which was capable of proving properties of computation in general, and in particular of proving the uncomputability of the halting problem [5].

### 1.2. Data Complexity

The first known attempts to quantify information were made by Nyquist (1924) [9] and Hartley (1928) [10]. Later, Shannon improved the information theory field by creating a quantitative model of communication as a statistical process underlying information theory. He improved upon the work of Nyquist and Hartley by defining the notion of average information (also called Shannon entropy) as the summation of the product between the probability of each character and the logarithm of this probability [11]. Moreover, he defined the basic unit of information, a bit, as the information entropy of a binary random variable that is 0 or 1 with equal probability, or the information that is obtained when the value of such a variable becomes known [12].

Following the work of Shannon, Solomonoff in 1960 presented the basic ideas of algorithmic information theory (AIT) as a method to overcome problems associated with the application of Bayes’s rule in statistics [13]. Later on, in a two-part paper published in 1964 [14,15], he introduced the notion of complexity (now broadly known as *Kolmogorov complexity*) as an auxiliary concept to obtain a universal a priori probability, and proved the invariance theorem governing AIT. This universal a priori probability M(x) is defined as the probability that the output of a universal Turing machine *U* starts with the string *x* when provided with fair coin flips on the input tape. M(x) can be used as a universal sequence predictor that outperforms (in a certain sense) all other predictors.

Algorithmic information theory was later developed independently by Kolmogorov and Chaitin, in 1965 and 1966, respectively. Kolmogorov proposed an improvement upon Shannon’s (probabilistic) description of information, adding the description of two more quantitative definitions of information–combinatorial and algorithmic [16]. Chaitin, besides the algorithmic nature [17], introduced Ω [18,19], a non-computable number [19], defined as the probability that an optimal computer halts, where the optimal computer is a universal decoding algorithm used to define the notion of program-size complexity. Chaitin also made substantial work regarding the halting problem [20,21,22].

In essence, Solomonoff, Kolmogorov, and Chaitin showed that, among all the algorithms that decode strings from their codes, there is an optimal one. This algorithm, for all strings, allows codes as short as allowed by any other, up to an additive constant that depends on the algorithms but not on the strings themselves. Concretely, algorithmic information is a measure that quantifies the information of a string *s* by determining its complexity K(s) (Equation (Equation 1)), which is defined by a shortest length *l* of the binary program *p* that computes *s* on a universal Turing machine *U* and halts [16],
(1)$$K\left(s\right):=\min_{p}\left\{l\left(p\right):U\left(p\right)=s\right\}.$$

This notion of *algorithmic Kolmogorov complexity* became widely adopted and is currently the standard to perform information quantification. It differs from Shannon’s entropy because it considers that the source creates structures which follow algorithmic schemes [23,24], rather than perceiving the machine as generating symbols from a probabilistic function. While Solomonoff applied this idea to define a *universal probability* of a string on which inductive inference of the succeeding digits of the string can be created, Kolmogorov used it to define several functions of strings, such as complexity, randomness, and information.

There are several variants of algorithmic complexity, the most successful of which was the prefix complexity introduced by Levin [25], Gács [26], and Chaitin [18]. Furthermore, an axiomatic approach to Kolmogorov complexity based on Blum axioms [27] was introduced by Burgin [28].

From the roots of algorithmic Kolmogorov complexity [29], Wallace formulated [30] the idea of minimum message length (MML) as compressed two-part codes for the data corresponding to replacing negative-log probabilities in Bayes’ rule by Shannon–Fano code lengths [31].

Independently from Wallace, Rissanenn described the minimum description length (MDL) principle in a series of papers starting with his paper in 1978 [32]. The basic idea underlying the MDL principle is that statistical inference can be used as an attempt to find regularity in the data. MDL is based on two main pillars: regularity implies higher compression and compression can be considered learning. Any regularity in the data can be used to compress the data; as such, regularity on data may be identified with the ability to compress them. On the other hand, compression can be viewed as a way of learning since the more we are able to reduce without loss the description of the data, the more we understand about its underlying structure. MDL joins these two insights and tells us that, for a given set of hypotheses *H* and dataset *D*, we should try to find the hypothesis or combination of hypotheses in *H* that compresses *D* the most.

Another possible way to quantify information was given by Bennett, who introduced the notion of Logical Depth. Essentially, it adds to Kolmogorov complexity the notion of time and, as such, can be defined as the time required by a standard universal TM *U* to generate a given string from an input that is algorithmically random [33].

An area strongly connected with AIT is the theory of algorithmic randomness, which studies random individual elements in sample spaces, mostly the set of all infinite binary sequences, e.g., a sequence of coin tosses, represented as a binary string. While Kolmogorov’s formalization of classical probability theory assigns probabilities to sets of outcomes and determines how to calculate with such probabilities, it does not distinguish between individual random and non-random elements. For instance, in a uniform distribution, a sequence of *n* zeros has the same probability as any other outcome of *n* coin tosses, namely $2^{-n}$. However, there is an intrinsic notion that such sequences are not random, which is even more pronounced in infinite sequences. The current view of algorithmic randomness proposes three paradigms to determine what a random object is: unpredictability (in a random sequence, it is difficult to predict the next element of a sequence); incompressibility (a random sequence is not feasibly compressible); and measure theoretical typicalness (a random sequence passes all feasible statistical tests). Martin-Löf in 1966 [34] provided a definition of a random sequence based on the measure of theoretical typicalness. Earlier researchers such as Von Mises [35] gave insights into the definition of a random sequence by formalizing a test for randomness. If a sequence passed all tests for randomness, it would be a random sequence. The problem was that, if we considered all possible tests, any sequence would not be random because it would be rejected by the test whose sole purpose was to reject that particular sequence. Martin-Löf’s key insight was to use the theory of computation to formally define the notion of a test for randomness. Namely, he did not consider the set of all tests but the set of effective tests (computable tests). This set of all effective tests for randomness can be subsumed by a universal test for randomness [34]. Unfortunately, it is not decidable by computation if a sequence passes the test. Although it is unfeasible in practice to apply the universal test for randomness, this notion can be approximated. An example can be found in compressors, since, by considering a string to be random and applying a series of models to compress the string, it is possible, although in a very limited way, to evaluate the randomness of the string. This is because, if a string is well compressed, there are repetitive patterns on the string, meaning it is not random. It is important to mention that other authors have made approaches to randomness via incompressibility (e.g., prefix-free Kolmogorov complexity [18,25,26]) or unpredictability (Constructive martingales [36]).

Despite the progress in the field of information theory, quantifying information is still one of the hardest open questions in computer science, since there is no computable measure that encapsulates all concepts surrounding information. Thus, one usually chooses between two options in order to quantify information: *Shannon entropy* or an approximation of *algorithmic complexity*.

Shannon entropy poses some problems since it is not invariant to the description of the object and its probability distribution [37]. Furthermore, it lacks an invariance theorem, forcing us to decide on a characteristic shared by the objects of interest [38]. On the other hand, *algorithmic complexity* is only approximately attainable, since the Kolmogorov complexity is non-computable [29]. These approximations are computable variants of the Kolmogorov complexity and are bounded by time and resources. Approximations were made by data compressors, as the bit stream produced by a lossless data compression algorithm allows the reconstruction of the original data with the appropriate decoder, and therefore can be seen as an upper bound of the algorithmic complexity of the sequence [39]. However, the problem of using fast general-purpose data compressors is that the majority of their implementations is based on estimations of entropy [40] and thus are no more related to algorithmic complexity than to Shannon entropy. There are however compressors which are embedded with algorithms, such as the DNA data compressor GeCo [41]. The role of specialized algorithms for specific data shows the importance of efficiently combining models that address both statistical and algorithmic nature. Although the statistical nature may be explored in advance, the algorithmic modeling requires exhaustive search, human knowledge, or machine learning.

Another substantial result was the algorithm proposed by Levin for solving inversion problems [42,43]. The Universal Levin Search (US) consists of an iterative search algorithm that interleaves the execution of all possible programs on a universal Turing machine *U*, sharing computation time equally among them, until one of the executed programs manages to solve the inversion problem provided. This search makes use of Levin complexity, which is a resource-bounded generalization of the algorithmic Kolmogorov complexity making it computable. In the context of a Turing machine, these resources are the maximum numbers of cells of the work tape used (space) and the number of execution steps (time). As such, a time-bounded version of Equation (Equation 1) can be obtained as in Equation (Equation 2), where it is considered the time taken to generate the string *s* executing all possible programs in lexicographic order [29],
(2)$$K_{t}\left(s\right):=\min_{p}\left\{l\left(p\right)+\log\left(\mathrm{time}\left(p\right)\right):U\left(p\right)=s\right\}.$$

Despite this, US is currently intractable due to hidden multiplicative constants in the running time and the need to make verification fast. To bridge this gap, Hutter proposed a more general asymptotically fastest algorithm, which solves these problems at the cost of an even larger additive constant [44,45].

More recent approaches such as the Coding Theorem Method (CTM) [46] and its derivation, the Block Decomposition Method (BDM) [38], approximate local estimations of algorithmic complexity providing a closer connection to the algorithmic nature. The main idea is to decompose the quantification of complexity for segmented regions using small Turing machines [46]. For modeling the statistical nature, such as noise, it commutes into a Shannon entropy quantification. This approach has shown promising results in different applications [47,48,49].

In this article, we study statistical patterns of specific TMs, with equivalent conditions, through global and local measures. For the purpose, we fix the source exclusively to an algorithmic nature although combining different rule configurations while keeping fixed the number of states and cardinality. Then, we measure and analyze the statistical complexity of the TMs tapes using compression-based approaches approximated by the best-order Markov model. We assess this compression-based approach according to different levels of symbol substitutions and permutations of contiguous blocks of symbols. After the definition and assessment, we identify some patterns in these TMs. We define and introduce the normal and dynamic complexity profiles as local measures to quantify and localize higher and lower regions of statistical complexity. We compare the BDM with our compression-based approach. Finally, we describe a simple algorithm to increase the statistical complexity of the tape while maintaining the same amount of algorithmic complexity.

The remaining of this paper is organized as follows. We start by presenting (Section 2) and validating (Section 3) our proposed method. In the following sections, we present the core results (Section 4), make a discussion (Section 5), and draw some conclusions (Section 6).

## 2. Methods

### 2.1. Turing Machines Configuration

In this study, we analyzed the output of simple TMs. All TMs have a binary or ternary set of symbols $\Sigma_{2}=\left\{0,1\right\}\;or\;\Sigma_{3}=\left\{0,1,2\right\}$, with a matching set of tape symbols Γ={0,1}orΓ={0,1,2}. These TMs have the following specific conditions:the machines start with a *blank* tape (all set to zero, b=“0”);the machines do not have an internal condition to halt (F=∅);the halting is performed by a external condition representing a certain number of iterations that is set equal to every single machine; andthe TMs are restricted to read only one tape character at a time and perform three types of movement on the tape, namely move one cell to the left, move one cell to the right, or stay on the same position.

Table 1 shows a matrix rule *M* example for a TM with #Q=2 and #Σ=2. Given a scanned symbol and the internal state of the TM, there is a triple that provides the TM information regarding the next symbol to write on the tape from the set $\Sigma_{2}$, the next internal state from the set Q={0,1}, and the next movement of the head relative to the tape (left = 0, stay = 1, right = 2). This matrix table is used to fill the tape across a given number of iterations. Each TM starts with the initial state of “0”, and the tape with the initial symbol “0”. With these initial conditions, the TM reads the tape’s symbol and, given the value read and the rule matrix, changes the tape symbol at the current position, its internal state, and the position of the head on the tape.

### 2.2. Search Approaches

Since we want to study the statistical complexity of TM tapes for different configurations, we created a large number of TMs, as specified in Section 2.1 with a small cardinality of states and alphabet (#Q,#Σ). These TMs were then analyzed as a whole, followed by a more particular analysis targeting TMs with specific configurations. For #Q×#Σ≤6, we performed a sequential search through all possible TMs. Since the total number of Turing machines (TNTM) increases in a super-exponential way (${\left(3\times \#\Sigma \times \#Q\right)}^{\#\Sigma \times \#Q}$), it becomes computationally intractable to run all the TMs for larger values of (#Q,#Σ) . Therefore, to approximate the results of a complete traversal of the TMs domain, we applied a Monte Carlo algorithm [50] to select TMs from their total pool for #Q∈{4,…,10} and #Σ=2, since this algorithm is widely used to obtain qualitative information regarding the behavior of large systems [51].

To perform a sequential search through all possible TMs of a pair (#Q,#Σ), we consider that each TM is represented by its rule matrix *M* and defined a *total order relation* between them. Furthermore, we considered this order to start with an initial TM ($tm_{0}$), which has its rule matrix filled with the same tuple (0,0,0). The definition of next(M)→M was based on a succession of increments which would carry to the next significant attribute on overflow. When the last possible *M* is reached, *M* returns to its configuration of $tm_{0}$. As an example, for #Q=#Σ=2, all four elements of the first TM rule matrix would start at (0,0,0), and the last TM would have all elements defined as (1,2,1).

To provide a unique numerical identifier (id) for each TM within the set of TNTM for a pair (#Q,#Σ), we mapped M→id (Algorithm A1 in Section A.1). The inverse function id→M is well defined and can be inferred from the former algorithm.

In contrast, in the Monte Carlo approach, rather than starting in one rule matrix and ending in another, TMs are randomly configured by sampling the components (write, move, state) of each cell from uniform distributions.

In both search cases, we took advantage of multiple processing cores of the same machine by subdividing the full group of TMs into separate jobs. For the sequential search, the domain was evenly split into as many partitions as the number of jobs in parallel. This separation was not required in the Monte Carlo algorithm. At the end of each job, the extracted measures were collected together into a single measure set.

### 2.3. Statistical Complexity

We used data compression to measure statistical complexity of the output tape created by the TM after *n* iterations. The Normalized Compression (NC) was used as the compression measure (Equation (Equation 3)) where *x* is the string, C(x) represents the number of bits needed by the lossless compression program to represent the string *x*, |A| is the size of the alphabet (equal to #Σ), and |x| is the length of the string *x*[52]. Furthermore, C(x) is computed as the probability *P* of each string symbol $x_{i}$ occurring,
(3)$$\begin{array}{c} NC\left(x\right)=\frac{C(x)}{\left|x\right|\times \log_{2}\left|A\right|} \end{array}\;,\mathrm{where}\;\begin{array}{c} C\left(x\right)=\sum_{i=1}^{\left|x\right|}-\log_{2}P(x_{i}). \end{array}$$

Probability P is computed using a Markov model, which is a finite-context model that predicts the next outcomes given a past context *k*[53]. Specifically, a Markov model loads the input using a given context *k* and updates its internal model. This internal model is used to compute the probability of any character being read at a given point. In the case of Algorithm 1, the tape produced by the TM is provided to the Markov model with context *k*. This model is used to determine the NC by computing the normalized summation of the probability of each character occurring on the tape.
**Algorithm 1:** Determine the NC of a generated TM tape.
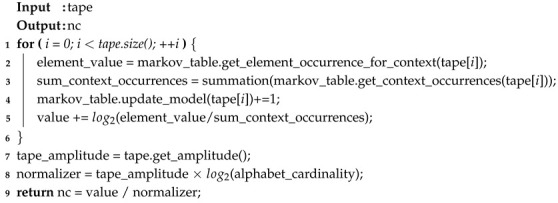


### 2.4. Normal and Dynamic Complexity Profiles

Some of the more statistically complex tapes obtained were analyzed individually by studying how the NC behaved with the increase in the number of characters currently written on the tape (amplitude of the tape). This study was carried out in two forms: a normal complexity profile and a dynamic complexity profile of the tapes. The normal complexity profile is computed after the TM is halted by an external condition representing a certain number of iterations, while the dynamic complexity profile is computed during the TM execution. It is also worth mentioning that the normal complexity profile was also applied to the sequence of rules used by the TMs.

A normal complexity profile can be seen as a numerical sequence $\vec{N}\left(x_{i}\right)$ containing values that express the predictability of each element from *x* given a compression function C(x). Assuming $x_{a}^{b}$ is a subsequence of *x* from position *a* to *b*, we define a complexity profile as:(4)$$\vec{N}\left(x_{i}\right)=C\left(x_{i}|x_{1}^{i-1}\right).$$

Notice that C(x) has a causal effect, which means that it is assumed that, for $\vec{N}\left(x_{i}\right)$, we have to previously access the elements $\vec{N}\left(x_{1}\right),\vec{N}\left(x_{\ldots }\right),\vec{N}\left(x_{i-1}\right)$ by order. To compare tapes with different alphabets, the profile was normalized according to
(5)$$\vec{C}\left(x_{i}\right)=\frac{\vec{N}\left(x_{i}\right)}{\log_{2}\left|A\right|},$$
where $\log_{2}\left|A\right|$ is the normalization factor by the cardinality of the alphabet.

The number of bits needed to describe *x* can be computed as the sum of the number of bits of each $x_{i}$, namely,
(6)$$C\left(x\right)=\sum_{i=1}^{\left|x\right|}\vec{N}\left(x_{i}\right),$$
where, as *i* increases, the compressor is asymptotically able to accurately predict the following outcomes, because it creates an internal model of the data. In other words, *C* is memorizing and, in some cases, learning.

The dynamic complexity profile can be defined as:(7)$$\vec{D}\left(x_{t}\right)=NC\left(x_{1}^{i},t\right),$$
where $x_{1}^{i}$ is the *x* sequence generated at the time iteration *t* considering that *x* is described from position 1 to *i*. As such, the dynamic profile was computed by providing the tape to the Markov model during the TM’s execution time (while the tape is being edited) and in small intervals the NC was computed as described in Section 2.3.

### 2.5. Increasing the Statistical Complexity of TM’s Tape

We also investigated the formulation of a methodology capable of consistently increasing the statistical complexity of the TM tapes, while maintaining the algorithmic complexity of the machines. To this end, we created two methods:Method I aims to increase statistical complexity by optimizing the impact of the rules on the TM’s statistical complexity (aggregation of key rules).Method II aims to globally increase statistical complexity by iteratively changing the TM’s rules.

The first method provided interesting conclusions regarding the interaction between key rules, whereas the second proposed method consistently improves tape statistical complexity. It is worth mentioning that in every computation of the NC, the best Markov model was selected, and the set of TMs has specific conditions (specified in Section 2.1).

#### 2.5.1. Method I

The first method is described in Section A.2 and Algorithm A2. The main principle of this method is to maximize the relevance of a given rule on the TM, by computing an impact measure of the rule using the NC. This metric is then used as a selection criteria of rules, when merging two randomly generated TM.

The premise is that a rule can be a key in a certain sub-network of rules at a certain time, and that by maximizing impact metrics we would be joining key rules that were relevant to the source matrix, successively. Even if during the TM merging process certain rules lost relevance, since the value of their impact decreased, these rules would be replaced by others that would better fit the TM rule matrix.

#### 2.5.2. Method II

In contrast to the previous method, this simpler method iteratively attempts to increase the statistical complexity of the tape by changing TM rules. This method, described in Algorithm 2, looks at the global impact a rule has on the statistical complexity of the TM’s tape, and tries to increase the global NC.

The algorithm goes as follows: (1) While the amplitude of the TM is smaller than a limit (set as 100 here), generate a random rule matrix, run TM through *n* iterations and determine the amplitude of the generated tape. (2) Compute the NC of resulting TM using its tape. (3) For the number of iterations defined: (i) randomly select a matrix rule index; (ii) randomly change the value of the rule; (iii) run TM for *n* iterations; (iv) determine the amplitude and NC of the generated tape; and (v) if the resulting NC is higher than the previous maximum and the amplitude is higher than 100 characters, keep the new rule. (4) Retrieve the new TM.

We evaluated the algorithms and apply both types of profiles in Section 4. In the next section, we assess our compression-based method regarding noise (substitutions) and permutations in synthetic and real genomic data.
**Algorithm 2:** Method II: Pseudo-code algorithm.
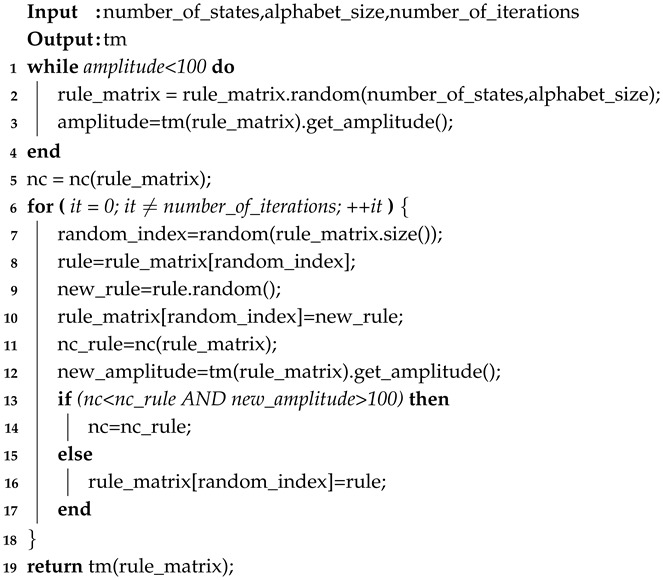


## 3. Assessment

In this subsection, we validate the use of the NC computed with the best Markov model. To make this assessment, we made use of synthetic and real inputs. The synthetic input was a string of 500 zeros followed by 500 ones, whereas the real inputs were the complete genome sequences of the *Microplitis demolitor bracovirus segment O* and the *Human parvovirus B19 isolate BX1* (both retrieved from https://www.ncbi.nlm.nih.gov/nuccore/). For each type of data, the substitution probability of the string was increased from 0% to 100% and, simultaneously, the string was randomly permuted in blocks of increasingly smaller sizes. At each point, the NC was computed using the best Markov model (k∈{2,…,9}) that minimizes the complexity quantity of that given string and the obtained results were plotted as a heat map (Figure 1). It is worth mentioning that, since the generated synthetic data have low statistical complexity and genomic data usually have high statistical complexity, in synthetic data, the substitution was randomly set to generate any letter of the alphabet Σ={0,1} in order to increase complexity of the string. On the other hand, in genomic data, the substitution was fixed to always generate a specific nucleotide (letter of the alphabet), in order to continually decrease the complexity.

As shown in Figure 1, the NC behaves as expected by, on the one hand, increasing as the substitution rate of the strings increases in synthetic data, and, on the other hand, decreasing in real data.

The same occurs regarding random permutation of the synthetic and natural strings. For the synthetic input, there was a significant increase of NC with the increase in number of blocks permuted. Furthermore, the same occurred in genomic data. Although less noticeable due to being more statistically complex than the synthetic sequence, in both genomic sequences, an increase in the NC with an increase in the number of blocks permuted can be seen. This is more pronounced in the *Microplitis demolitor bracovirus* genome sequence than in the *Human parvovirus B19 isolate BX1* genome sequence since the latter has a more statistically complex nature.

These results imply that the computation of the NC in this manner is both a sensitive and tolerant way of dealing with substitutions and permutations. As such, this methodology could be successfully applied to measure the output of TM tapes, since, besides being an ultra-fast method of obtaining information regarding the tape, it can cope with the presence of substitutions and permutations in a string and, thus, is a good estimator of statistical complexity on a tape.

## 4. Results

In this section, the statistical complexity of TMs with different alphabet and state cardinality (Section 4.1) is analysed. This is followed by the results obtained by applying the normal and the dynamic complexity profiles to statistically complex TMs (Section 4.2). Afterwards, our compression-based methodology is compared with the BDM (Section 4.3). Finally, the results of our methodologies for increasing the statistical complexity of TM are exposed (Section 4.4). All results can be reproduced, under a Linux OS, using the scripts provided at the repository https://github.com/asilab/TMCompression.

### 4.1. Statistical Complexity Patterns of Turing Machines

All TMs ran for 50,000 iterations, as this is a large number of iterations that can still be computed in reasonable time. The importance of having a large number of iterations results from the observation that increasing iterations led to an increase in the compression rate, which by definition implies that more redundant sequences were more easily compressed, whereas less redundant ones were not. The tapes produced by each TM were analyzed with Markov models of context k∈{2,…,9} and the smallest NC was selected. Table 2 shows the average amplitude of the tape and NC as well as its maximum standard deviation obtained for each group of TMs with a different pair of (#Q,#Σ). Due to limited availability of computational resources, the number of sampled TMs in the Monte Carlo search approach was the same order of magnitude as the TNTM for #Q=3 and #Σ=2.

A low-pass filter was applied to the obtained values of data compression, the tape’s amplitude was normalized by the maximum size obtained for its pair (#Q,#Σ), and the results were expressed as plots for #Q∈{2,…,6}, as presented in Figure 2. For higher cardinalities, plots were not created due to the under-sample of computed TM relative to the TNTM. The plots show an interesting relationship between the tape’s amplitude and the NC. In particular, the NC tends to be higher for tapes with a smaller amplitude. There are two major reasons for this: On the one hand, smaller strings are harder to compress, due to tape repetitions being poorly represented. This happens mostly in machines that only produce 1–10 elements in the tape. On the other hand, there is also a correlation between smaller tapes created by TMs with intertwined rules and a high NC value. This is caused by TMs re-writing the tape several times, thus editing portions of it.

Overall, TMs show two large regions where there is a large spike in NC and smaller amplitude of the tape. These regions persist for TMs with a different #Q and #Σ, at least for the cardinality under examination (see circles in Figure 2).

To examine this, we computed the average bits required to represent the generated tapes inside these regions, as well as the NC and the amplitude of the tape. Furthermore, we sampled from this pool 5000 TMs from inside and outside this region (except for #Q and #Σ=2, with only 2000 selected). This sampled TMs ran for 1000 iterations to obtain the list of the index of rules used by each machine. From this list, the NC and average bits required to represent the list of index rules were computed. Figure 3 depicts these results: (top-left) the amplitude of TM’s tape; (top-middle) the required bits to perform compression of the tape; (top-right) the corresponding NC value; (bottom-left) the average required bits to perform compression of index rules used by each TM; and (bottom-right) its the corresponding NC value.

By analyzing Figure 3, we observe that, for all TM(#Q,#Σ), the amplitude of the tape and bits required to compress it are on average higher outside the circle lines than inside this region. In contrast, the average NC is higher inside the region. This tells us that, despite the seemingly higher statistical complexity, tapes are in fact not more statistically complex but rather simply smaller in size. This can be concluded since on average the tapes inside this region require fewer bits to represent than those outside, meaning that the logarithmic normalization factor of the NC is creating this discrepancy in values and not the complexity of the sequence itself.

Regarding the NC and the average bits required to represent the index of the rules used, we see that in both cases the indexes of the rules are on average more easily compressible inside the regions under analysis (Figure 3, bottom). This indicates that there are smaller rule cycles in this regions, making it easier for the Markov models to compress it. To see this, we performed the normal complexity profile of these rules and computed its average. Figure 4 shows the important regions of these rule complexity profiles (all complete profiles are shown in Appendix B, Figure A1).

The rule complexity profiles have a higher complexity outside the region than inside. Furthermore, the decrease in complexity occurs faster inside the region than outside, being that this difference is incremented with the increase in cardinality of states. This demonstrates that the rule cycles are smaller inside this region. These small rule cycles (that create low statistical complexity) are going back and forth in very small regions of the tape, decreasing the overall size of the tape. This decrease in size coupled with the low complexity of the rules produces, on the one hand, models that require less bits to represent the tape, and, on the other hand, a higher NC value due to its dividing factor |x|.

To conclude, given the same number of iterations, TM tapes in these regions possess smaller amplitude and higher NC due to similar short cycles. The regions in Figure 2 were created because these short cycles occurred in TMs, which were grouped together due to the statistical complexity of the TM tapes being measured assuming a sequential generation order of the rules (changes in the rules sequential).

### 4.2. Normal and Dynamic Complexity Profiles

To create the normal and the dynamic complexity profiles, we filtered from each pair (#Q,#Σ) in Table 2 the top 15 TMs that had a tape amplitude larger than 100 characters and the highest NC. Figure 5 depict the normal and the dynamic complexity profiles respectively obtained from the TMs with highest NC for its state cardinality, when run until each has a tape amplitude of 10,000.

Regarding the normal complexity profile, we can make some conclusions. Firstly, all machines have tape regions which are harder to compress given the prior knowledge of the Markov table, thus the spikes. This indicates that there are regions where tape pattern has changed due to a change in state of the machine and/or part of the tape being re-written.

It is also worth mentioning that, similar to Figure 1 (**top**), where an increase in the number of string editions leads to an increase of the statistical complexity of the string, the profiles of Figure 5 (**left**) show that tape regions which have suffered more edition due to complex rule interchange are harder to compress (more statistically complex).

Regarding the dynamic complexity profile, TMs that create a tape with simple repetitive patterns have a dynamic complexity profile that decreases their NC across time, until it reaches approximately zero. This behavior is also observed in TMs with the highest NC for (#Q∈{2,3,4},#Σ=2). However, with an increase in the number of rules, the compression capacity of these tapes starts to decrease, due to more complex patterns being generated. These patterns change during the creation of the tape due to interlocked changes in TM states and changes on received inputs.

As expected, in both the normal and the dynamic complexity profiles, with the increase in #Q and #Σ, there is an increase in the number and size of spiky regions (normal complexity profile) and the value of NC (dynamic complexity profiles), indicating that the presence of more rules influences the amount of information present on the tape, thus making it generally harder to compress.

In general, these results, as well as those exposed in the previous subsection (normal complexity profiles applied to TM rules), show that these profile methods are capable of localizing higher statistical complexity for different purposes. The normal complexity profile is capable of localizing regions of higher and lower statistical complexity (assuming the machine reached the external halting condition), as well as localizing regions of small cycles of rules. On the other hand, the dynamic complexity profile is capable of localizing temporal dynamics of statistical complexity, showing where there is an abrupt change in statistical complexity.

### 4.3. Comparison between NC and BDM

Besides assessing the relationship between the algorithmic complexity of TMs and the statistical complexity produced by their tapes using the NC, as well as applying the normal and dynamic complexity profiles as local measures to quantify and localize higher and lower regions of statistical complexity, we also compared the NC and the BDM for assessing the statistical complexity of tapes. For that purpose, we pseudo-randomly selected 10,000 TMs with 6, 8, or 10 states and a binary alphabet, and ran each for 50,000 iterations. To the tapes obtained, the NC and the One-Dimensional normalized BDM were applied according to Zenil et al. [38]. The average results can be observed in Table 3, and the overall behavior for TMs with #Q=10 and #Σ=2 can be seen in Figure 6 (images for all analyzed states are shown in Figure A2 in Appendix C). Notice that, in Figure 6, the results are presented after applying a low-pass filter. Furthermore, tape amplitude was normalized by the maximum size obtained for its pair (#Q,#Σ) and, to observe BDM comparatively to the NC, in the left image the BDM results were scaled by a $10^{2}$ factor. An example with non-scaled BDM is also presented in the right image.

Similar to Table 2, there is a decrease in amplitude of the tape with the increase of the number of states. Additionally, in this sample, both BDM and NC increase with the increase in number of states. The results indicate that the algorithmic nature of BDM allows a very small representation of these data generated by an algorithm. This is because the generation of the tapes was created by Turing machines and BDM is constituted by a set of small Turing machines besides the Shannon entropy approximation. Nevertheless, for a higher number of states, the BDM is progressively approximated by our methodology.

### 4.4. Increase Statistical Complexity of Turing Machines Tapes

Methods I and II were implemented and tested regarding their capability to increase statistical complexity. Method I was instantiated 200 times, where, in each instance, 100 TMs were merged according to Algorithm A2. On the other hand, Method II was instantiated 2000 times (10 times more since it is a faster algorithm) for 1000 iterations. In both methods, all TMs had a #Q=10 and #Σ=2, and were executed 50,000 times. The compared results can be observed in Figure 7.

Looking firstly at Method I (Figure 7, left), both top and bottom plots show that this algorithm decreases the tapes amplitude to half, while retaining approximately the same bits required to represent the tape and slightly increasing the NC. This method shows inconclusive results regarding the increase in the statistical complexity of the tape. The bits required to compress the tape are on average approximately the same, whereas the amplitude of the tape decreases significantly, explaining the increase in NC. On the other hand, the tapes decrease by half on average, while retaining approximately the same number of bits, which could mean that information on the tapes is being condensed. As such, more work needs to be performed on the method in order to conclude its capability to increase the statistical complexity of TMs.

Moving on to the results obtained for Method II, both top and bottom plots show that this algorithm on average decreases tape amplitude while significantly increasing the bits required to represent the tape and the NC value. Besides being much faster than the previous method, it can systematically increase the statistical complexity of TMs. This happens because this method tries to change the rules by looking at the outcome of the TM’s tape NC, rather than augmenting the impact of the rule itself on the tape. Figure 8 presents two tape evolution examples using Method II.

To better evaluate Method II, we changed the number of rule iterations it uses to improve (it∈{50,100,200,…,32,000}), the number of iterations the TM can write on the tape (n∈{500,1000,2000,…,8000}), and computed for 2000 instances: (1) the final average amplitude of the tape; (2) the variation of the bits required (BR) to represent the tape $\left(\bar{{\mathrm{BR}}_{\mathrm{var}}}=\frac{1}{n}\times \sum_{i=1}^{n}\left(BR_{final}-BR_{initial}\right)\right)$; and (3) the variation of NC $\left(\bar{{\mathrm{NC}}_{\mathrm{var}}}=\frac{1}{n}\times \sum_{i=1}^{n}\left(NC_{final}-NC_{initial}\right)\right)$. The results of this process are shown in Figure 9.

It is possible to observe in Figure 9 that both rule iterations and iterations on the tape play a big role on the statistical complexity of the TM. As expected, the method demonstrates that, as the number of rule iterations increases (iterations that try to increase the NC value of the tape), both the variation of the NC (maximum average increase of 40%) and bits required (maximum average increase of 55 bits) increase, whereas the amplitude of the tape decreases (minimum of 111 characters of tape amplitude).

On the other hand, the variation of the NC and bits required act in a very interesting way. Firstly, for a reduced number of rule iterations, the NC is higher for a lower number of tape iterations, showing that there is an overall improvement of the NC due to the small tape size. However, as the number of rule iterations increases, so does the variation of the NC and bits required, showing that there is in fact a solid increase in the statistical complexity of the TMs’ tapes. This means that the sequence is getting increasingly more complex with the number of iterations, since the only way to sustain an increase in complexity is by creating rules with non-repeatable patterns. As such, the decrease in tape amplitude observed when using Method II implies that the rules obligate the rewriting of certain parts of the tape, making it more compact and thus harder to describe by a statistical compressor. A simple adaptation of Method II can be done to achieve a decrease of statistical complexity while maintaining the same algorithmic complexity (the opposite). This allows the variation of the statistical complexity of a TM tape according to a desired value, while maintaining the same algorithmic complexity.

## 5. Discussion

In this study, we analyzed the statistical complexity of Turing machine tapes with specific conditions. The conditions of the machines are: the initial tape is a *blank* tape (all set to zero), the machines do not have an internal condition to halt, and the halting is set by an external condition representing a certain number of iterations that is set equal to every single machine.

Fixing the algorithmic complexity using the same number of states and tape cardinalities, we were able to measure the statistical complexity of the tapes by a compression-based algorithm that finds the best-order Markov model, contained in a set of available contexts. We benchmark this fast compression-based approach, showing that it is tolerant to high levels of substitutions of symbols and permutations of contiguous blocks of symbols.

Using this compression-based approach, we introduce the normal complexity profiles and dynamic complexity profiles as approximate measures that can detect complexity change in the events through spatial and temporal quantities. The normal complexity profiles localize regions of higher and lower statistical complexity assuming the machine reached the external halting condition while the dynamic complexity profiles localize temporal dynamics of statistical complexity through the dynamics of the tape (while running). The latter identifies when a change in complexity occurs at a certain depth. It is worth mentioning that a similar concept, Algorithmic Information Dynamics (AID), was introduced by the Zenil, Kiani, and Tegnnér [54]. AID consists of exploring the effects that perturbations to systems and data (such as strings or networks) have on their underlying computable models. It uses algorithmic probability between variables to determine probability for the dependency (and direction between variables) as well as the set of candidate models explaining such a connection. On the other hand, the dynamic complexity profiles localize temporal dynamics of statistical complexity through the modification cycles of the tape. Namely, it shows where there is abrupt change in the statistical complexity during the TMs execution. It detects the rules that cause direct influence on statistical complexity of the tape during the TMs run time. Although these notions are similar, they differ in the fact that one relies on NC to compute the statistical complexity of the tape in a dynamic temporal way, whereas the other uses algorithmic probability to determine the smallest program to represent that pattern after the perturbations have taken place.

Using the Normalized Compression, we showed that some Turing machines (with the mentioned conditions) have higher statistical complexity (and lower tape amplitude) in two regions (shown up to six states). We localized these regions and, through the complexity profiles, identified the cause as short cycles in the rules that output tapes with lower amplitude. Since the statistical complexity of the Turing tapes was measured assuming a sequential generation order of the rules (changes in the rules sequential), it groups these similar short cycles given their similar configurations.

Another contribution in this paper is a comparison with a measure that uses both algorithmic and statistical approaches (BDM) for analysis of the tapes. Naturally, given the generation of the tapes being created by Turing machines with the mentioned specific conditions, it is efficiently described by the BDM. Nevertheless, for a higher number of states, the BDM is progressively approximated by our methodology since the algorithmic approach of the BDM is constituted by a set of small Turing machines, currently, up to a maximum number of five states (binary alphabet). On the other hand, the statistical approach used in the BDM is more simplistic than ours, making the proposed method more relevant for this specific application. Therefore, since the purpose of this study is to quantify the statistical complexity of the TM tapes, the BDM is not suitable to be used in this case because the BDM goes beyond statistical complexity. Nevertheless, for an application of quantification of both algorithmic and statistical complexity, a combination of both methods would approximate the complexity of strings efficiently.

The final contribution of this paper is a simple algorithm for increasing the statistical complexity of the tape while maintaining the same amount of algorithmic complexity. The value of the statistical complexity can be set to the desired number instead of the maximum. This algorithm can be applied for ultra-fast generation, from a short program, outputting tapes with a desired statistical complexity while the algorithmic complexity remains low. This algorithm may have many applications in the field of genomics, metagenomics, and virology. For example, the output of these Turing machines can be used as an input to test and quantify the accuracy of genome assembly algorithms for different complexities with non-stationary sources assuming the statistical line (high-speed generation without the need to store any sequence besides the configurations). Another application is the simulation of progressive mutations in metagenomic communities. It can also be used to complement absent regions of genome viruses given sequencing incapabilities or DNA/RNA degradation (using similar statistical complexity). A broader application is the benchmark of compression algorithms that are used to localize complexity in the data given possible algorithmic sources.

Measuring the output of Turing machine tapes requires ultra-fast methods. The majority of lossless compression algorithms is limited to finding simple statistical regularities because these algorithms have been designed for fast storage reduction. However, there are some which are designed for compression ratio improvement at the expense of more computational resources. For example, lossless compression algorithms, such as GeCo [41], are hybrids between statistical and algorithmic schemes. In fact, GeCo, besides context models, uses sub-programs of substitution [55] and inverted repeats [56] for efficient representation of DNA sequences. Another example is PAQ8 (http://mattmahoney.net/dc/), a general-purpose compressor, which combines multiple context models using a neural network, transform functions, secondary structure predictors, and other light sub-programs. This algorithmic approach differs from the majority of the lossless compression algorithms since they consider that the source creates structures that follow algorithmic schemes, and, as such, do not only perceive data as being generated from a probabilistic function. However, these compressors are programmed by humans without taking into account all the algorithmic nature present in the data. Nevertheless, these compressors show the importance of using data-specialized algorithms as well as the importance of efficiently combining models that address both statistical and algorithmic nature.

Although the statistical nature of compressors is highly studied, algorithmic modeling is yet to be fully explored. Usually, the problem is how to find fast and efficient algorithmic models for data compression. We know two approaches: human inspection and automatic computational search. Human inspection is, mostly, subjective, unpredictable, and requires programming time, while automatic computation is incremental and constant. In a sense, a small difference can make all the difference. Indeed, lossless data compressors are tightly related to the concept of minimal description length [32] and a shortest program to represent a string and halt [29,57,58]; therefore, representative algorithms can be efficiently embedded in these data compressors. On the other hand, the characteristics and quantity of noise, precision, and intention of most outputs from nature, on average, inverts the efficiency of programs to measure the information that is based on pure algorithmic schemes than those which have a more probabilistic nature. Nature can be based on algorithmic schemes, but is not limited to them. In this case, we believe that another part is based on non-computable sources that make logical descriptions inconsistent [59,60].

In this study, we evaluated the usage of the Normalized Compression (NC) computing the selection of the best Markov model that minimizes the complexity quantity of the Turing machine tape. Although the algorithmic complexity of the TMs that generate the tapes is the same for a set of (#Q,#Σ) since the number of rules is the same, the output tapes can vary widely in size and form (a good example being the Busy Beaver) and thus the approximation of the statistical complexity provided by the NC of the tapes generated differs substantially from tape to tape. While it is relatively easy to compress repeated sequences, the compression of sequences with an algorithmic scheme has proven difficult to do by the simple statistical compressor. However, it is simple to describe using TM’s rule matrix. This is an inversion problem since the problem is how to obtain the rule matrix given the produced TM’s tape. It is intractable to perform a sequential search due to the TNTM following a super-exponential behavior. One might try to tackle the problem with a resource-bounded approach similarly to Levin [42,43] or Hutter search [44,45], although in search of long strings the problem also becomes unfeasible. More recent approaches are the Coding Theorem Method (CTM) [46] and its derivation, the Block Decomposition Method [38], which decompose the quantification of complexity for segmented regions using small Turing machines, or modeling the statistical nature, such as noise, it commutes into a Shannon entropy quantification. This approach is a numerical method that implements US in a resource-bounded way and has shown promising results in different applications. According to MDL, compression can be viewed as a way of learning since the more we can reduce data without losing their description, the more we understand about their underlying structure [32]. As such, by combining both statistical compressors and algorithmic compressors, one might better describe data and sequentially understand its underlying nature. Our studies seem to indicate that a simple way to do this might be to make use of simple statistical compressors to identify different regions in data (more probabilistic scheme regions and more algorithmic ones) and to those regions apply more appropriate methods. In other words, apply optimal statistic data compressors to parts of data that have a more probabilistic nature (easily compressible by the simple statistical compressor) and apply compressors with an algorithmic nature to parts of data that follow algorithmic schemes (such as the rules of the TM that better represent that region). If it is possible to come up with a close algorithmic representation of the digital object, the applications are tremendous, since we can (besides losslessly representing data in a compressed manner), get insights regarding the underlying structure of data itself. In practice, this could be used in genomics to determine the algorithmic nature of the sequences and predict their behavior, and in medicine as a tool to perform predictive diagnosis.

## 6. Conclusions

Turing machine tapes with the same algorithmic complexity may have very different statistical complexities. In this study, we analyzed the behavior of TMs with specific conditions regarding the statistical complexity of their generated tapes. To carry out this study, we created low (algorithmic) complexity TMs (a small number of states and alphabet) and used a compression-based measure approximated by the best-order Markov model to measure the quantity of statistical complexity in the output tapes.

After assessing the compression-based approach regarding symbol editions and block transformations, we identified that TMs with the same number of states and alphabet produce very distinct tapes with higher complexity patterns in two regions. We showed that these two regions are related to repetitive short cycles in the configuration matrix, given sequential modifications in the matrix that create shorter tapes.

We introduced normal and dynamic complexity profiles as approximate measures to quantify local complexity. The normal complexity profiles localize regions of higher and lower statistical complexity for machines that reached the external halting condition. The dynamic complexity profiles localize temporal dynamics of statistical complexity through the modifications cycles of the tape. We showed examples using both, and applied them to achieve some conclusions in this article regarding global quantities.

Additionally, we made a comparison with a measure that uses both algorithmic and statistical approaches (BDM) for analysis of the tapes. Naturally, given the generation of the tapes being created by Turing machines with the mentioned specific conditions, it is efficiently described by the BDM. However, for a higher number of states, BDM is progressively approximated by our methodology.

Finally, we developed an algorithm to progressively increase the statistical complexity of a tape. This algorithm creates a tape that evolves through a stochastic optimization rule matrix that is modified according to the quantification of the Normalized Compression. We pointed out some of the applications of this algorithm in the bioinformatics field.

## Figures and Tables

**Figure 1 entropy-22-00105-f001:**
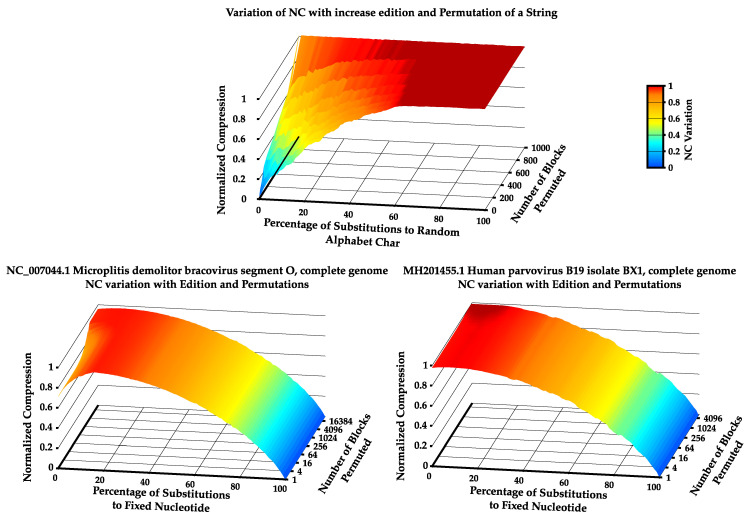
Heat map of Normalized Compression with an increase in permutation and edition rate. Generated string starting with 500 zeros followed by 500 ones (**top**); NC_007044.1 *Microplitis demolitor bracovirus segment O*, complete genome (**bottom-left**); and MH201455.1 *Human parvovirus B19 isolate BX1*, complete genome (**bottom-right**).

**Figure 2 entropy-22-00105-f002:**
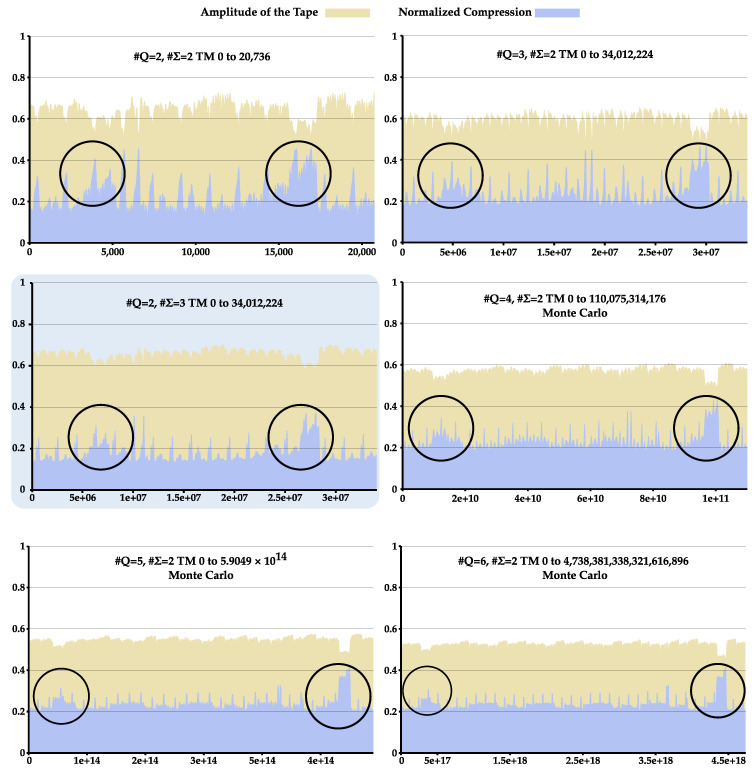
Plot of all TMs in Table 2. NC value is in blue and the tape’s normalized amplitude size is in yellow. The x-axes of the plots represent the index of the Turing machine computed according to Algorithm A1. The **blue background** is the plot that corresponds to the group of TMs with #Σ=3 and #Q=2; all other plots have #Σ=2.

**Figure 3 entropy-22-00105-f003:**
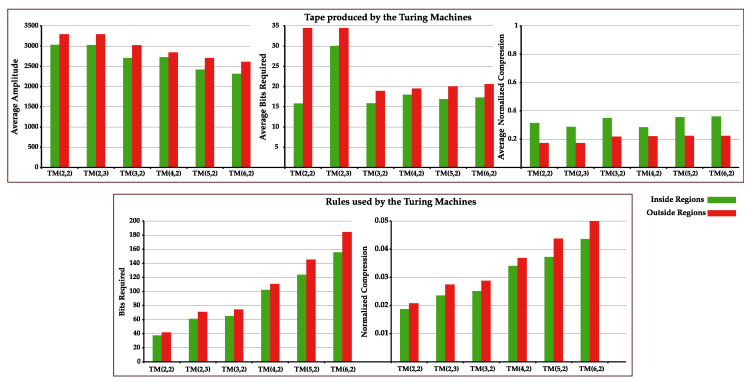
The average value for the amplitude of TM’s tape (**top-left**); average required bits to perform compression of the tape (**top-middle**); and average NC value (**top-right**), inside and outside the regions marked with circles in Figure 2. The average bits required (**bottom-left**); and the average NC value obtained for the rules used by the TM (**bottom-right**), inside and outside the regions marked with circles in Figure 2

**Figure 4 entropy-22-00105-f004:**
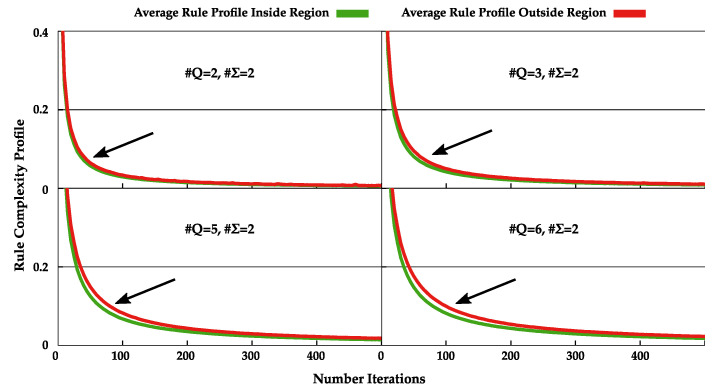
Regional capture of average rule complexity profiles obtained from pseudo-randomly selected TMs with #Q∈{2,3,5,6} and #Σ=2 up to 1000 iterations.

**Figure 5 entropy-22-00105-f005:**
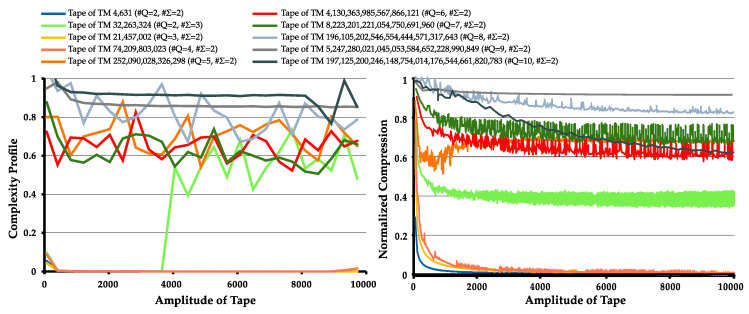
Normal complexity profiles (**left**); and dynamic complexity profiles (**right**) obtained for some of the filtered TMs. Each TM has a different cardinality of states or alphabet.

**Figure 6 entropy-22-00105-f006:**
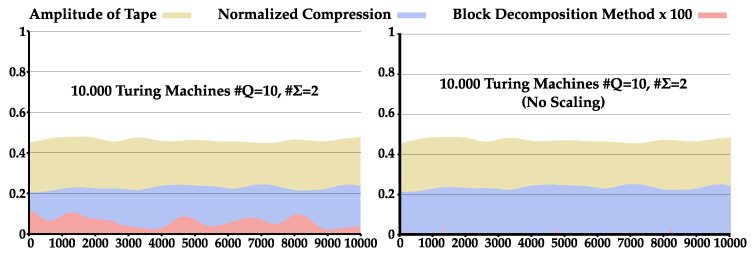
Comparison between the NC and BDM for 10,000 TM with #Q=10 and #Σ=2 that ran over 50,000 iterations: (**Left**) BDM scaled by a factor of $10^{2}$; and (**Right**) same example but with non-scaled BDM.

**Figure 7 entropy-22-00105-f007:**
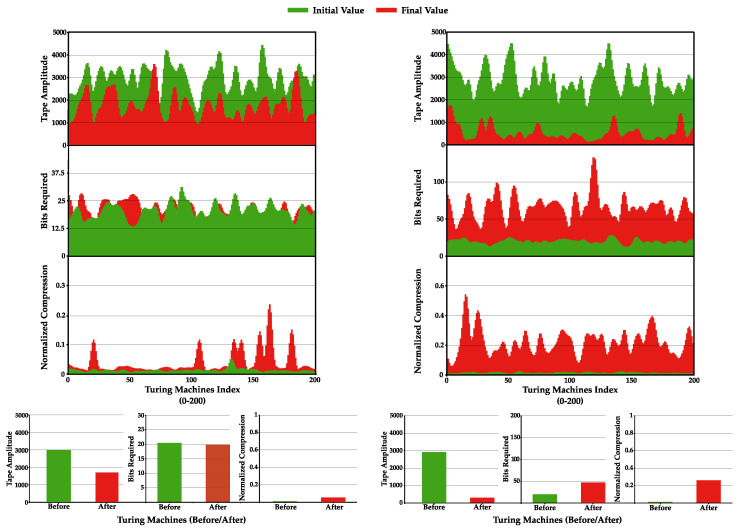
Comparison of: Method I (**left**); and Method II (**right**). (**Top**) Plots show the amplitude of the tapes, bits required to represent the sequence, and the NC obtained for 200 TMs after a low-pass filter was applied. (**Bottom**) Plots show the average tape amplitude (**bottom-left**); average bits required (**bottom-middle**); and NC (**bottom-right**). Green and red colors represent TMs before and after the method was applied, respectively. For Method I, the average corresponds to 200 instances and for Method II to 2000.

**Figure 8 entropy-22-00105-f008:**
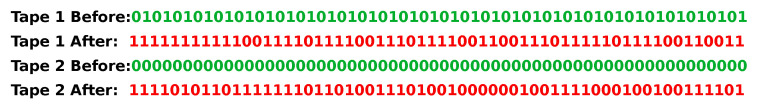
First 59 characters of TMs’ tapes before and after the Method II was applied.

**Figure 9 entropy-22-00105-f009:**
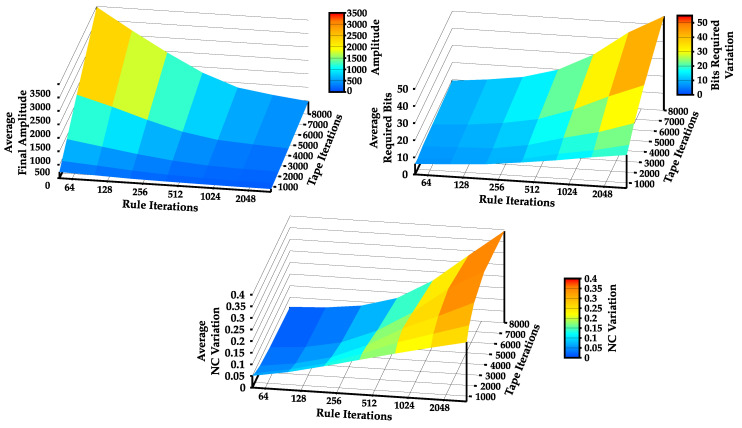
Average final amplitude of the tape (**top-left**); variation of the bits required to represent the string (**top-right**); and variation of the NC (**bottom**), with the increase in number of rule iterations and tape iterations.

**Table 1 entropy-22-00105-t001:**
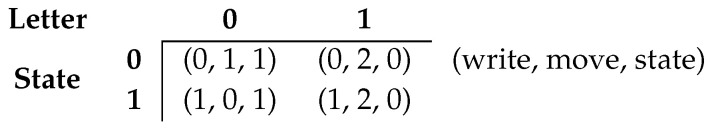
Rule matrix for a TM with #Q=2 and #Σ=2.

**Table 2 entropy-22-00105-t002:** The average and maximum standard deviation of the amplitude of the tape and NC obtained in each TM group.

#Σ	#Q	TM No.	Search Approach	Mean Amp ± std	Mean NC ± std
2	2	20,736	Sequential	32,055 ± 20,836	0.22724 ± 0.42095
2	3	34,012,224	Sequential	29,745 ± 20,609	0.22808 ± 0.42330
3	2	34,012,224	Sequential	32,603 ± 19,923	0.17887 ± 0.38230
2	4	34,000,000	Monte Carlo	28,146 ± 20,348	0.22753 ± 0.42403
2	5	34,000,000	Monte Carlo	26,932 ± 20,092	0.22643 ± 0.42403
2	6	50,000,000	Monte Carlo	25,963 ± 19,856	0.22512 ± 0.42363
2	7	50,000,000	Monte Carlo	25,164 ± 19,636	0.22356 ± 0.42285
2	8	30,000,000	Monte Carlo	24,477 ± 19,433	0.22219 ± 0.42215
2	9	30,000,000	Monte Carlo	23,882 ± 19,245	0.22079 ± 0.42134
2	10	30,000,000	Monte Carlo	23,357 ± 19,068	0.21933 ± 0.42039

**Table 3 entropy-22-00105-t003:** The average and maximum standard deviation of the NC and NBDM obtained in each TM group.

#Σ	#Q	TM No.	Mean Amp ± std	Mean BDM ± std	Mean NC ± std
2	6	10,000	26,359.6 ± 19,821	0.00042801 ± 0.011328	0.21631 ± 0.41753
2	8	10,000	24,471.8 ± 19,353	0.00045401 ± 0.010250	0.21975 ± 0.42054
2	10	10,000	23,123 ± 19,157	0.00060360 ± 0.012539	0.22696 ± 0.42404

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
