# Peer review of "Statistical Complexity Analysis of Turing Machine tapes with Fixed Algorithmic Complexity Using the Best-Order Markov Model"

_entropy, 2020, doi:10.3390/e22010105_

Round 1

Reviewer 1 Report

1. Content

The objective of this paper is nicely summarised in the first paragraph of its conclusion: The authors have aimed to analyse the behaviour of very small Turing machines regarding the question of how "complex" the tapes contents will become when these machines are run on an initially empty tape. The hypothesis is that even some very small Turing machines should be able to generate tape contents that are complex from a statistical point of view.

This is the second time I got to review the paper. Compared to the first version, the presentation is still very nice, especially the introduction, but there has now been a noticable improvement regarding the question of what the actual contributions of the paper are:

- The authors "actually did the analysis" in the sense that this has most likely not been done before (at least not in a systematic way). The paper includes many colourful graphics and plots showing how the statistical complexity of Turing machine tapes changes when we vary the transition function of the machine, the number of iterations, or a number of other parameters.

- The authors draw some conclusions from their findings: First, the argue that a Method I they investigated to increase the complexity of the tape contents by a sort of evolutionary algorithm for finding transition functions "does not work" - and it is sometimes more important to know that an algorithm does *not* work than knowing that is does. Second, they argue that a Method II, which is a local search in the space of Turing machine transition tables, gives machines that produce tape contents of a higher statistical complexity that random ones. Third, they identify "areas and spikes in the space of transition tables that produce increased statistical complexity".

2. Presentation

As mentioned above, the presentation is pretty good in the sense that the paper is carefully written (I list some typos at the end of this review). In general, one can follow the arguments and understand what is going on.

Some general things that I think could be improved throughout are:

- The many plots have been created with external programs and been then scaled in different ways. This creates a rather inconsistent "look" with some figures containing text that is almost impossibly small to read. The fonts are also wrong. Well, I guess this is what most people do, but a bit more effort here might be nice.

- More importantly regarding the figures, I am not a fan of 3D plots since it is hard infer any actual datapoints from the 2D rendering. For instance, in Figure 1, anyone would be hard-pressed to say what is, in the the lower left plot, the NC value for 50% substitutions and 256 blocks. Something between 0.6 and 0.8, I guess. I know that these 3d plots "look fancy", but I the great statistician Edward Tufte would call them junk charts, I guess. At the very least, avoid the totally unnecessary extra rotation by 10 degrees that ensures that not a single axis is horizontal or vertical (which would at least make it possible to draw some inferences from the plot). In general, in visualising data, perspectives are evil and should be avoided whenever possible.

- Regarding the program code, please consider a serious overhaul of the presentation. The code is now a very weird mixture of C++ and pseudo-code. Stuff like "for (i=ruleMatrix.begin(); i \neq rulematrix.end(); ++i)" will be understandable only to someone fluent in C++ (like me...), but anyone else will have a hard time. It seems that the code has been copied directly from the code with even the abbreviated source code variable names being copied. Having programmed at least a million lines of code in my life, I can sympathise that one names a variable mrkvTable rather than Markov_table or tmMerge rather than Turing_Maschine_merge, but these abbreviations do not belong in a paper. Neither do C++ idioms like begin / end iterator stuff.

As a small final remark, while the introduction is nicely written, the beginning is what is known as an "Adam and Eve" beginning, tracing routes back to the earliest writing systems - I would not have been surprised to have the bible mentioned there. I suggest starting with Shannon...

3. Scientific contribution

Unfortunately, despite the nice presentation and the amount of work that has undoubtedly gone into this paper, I feel that the scientific contribution is somewhat meagre. While in the previous version of this paper I did not even see what the contribution was supposed to be, this is now clear - but now I sadly feel the contribution is small.

My basic criticism concerns the approach taken in the paper towards identifying Turing machines that have a high statistical complexity: The authors treat Turing machine tables as objects spanning a search space that they sample and in which their algorithms move around (Methods I and II). Especially Method I is based on the premise that the individual "rules" in the table are "better" than other one and might, thus, lead to a higher statistical complexity of the output. Sorry, but that feels like, well, nonsense, to me: Turing machine tables are computer programs by any other name. Trying "to determine whether rule (0,1,0) is better than (2,1,0)" is like trying "to determine whether the assignment x=5 always produces faster programs than the assignment y=3 in a program". This makes no sense and it is, thus, no coincidence that Method I fails. The fact that Method II achieves something is also not due to some rules being better than others, but rather due to the fact that a local search through a not totally random search space will produce some improvements. And it does.

I guess the (laudable) objective of the paper was to somehow get a grip on the question of how the very low complexity of small TMs produces "random-looking tape contents" and to quantify the amount of randomness that arises. However, despite the many experiments that authors did and the many plots in the paper, I do not see that any real insights into this question have been obtained.

One could run the same set of experiments and write almost the same paper again when one replaces Turing machines by cellular automata: Consider different rule sets and different starting positions and watch the beautifully complex tapes that arise. (Wolfram wrote a 1500 page book on this stuff...) One could write the paper once more for different starting values of a fractal. Or of a strange attractor, or of a chaotic system. And so on, and so on.

At least for me, it is not surprising that from a very small amount of information (a small transition table) we see highly complex outcomes (tapes) arising. The interesting part for Turing machines -- a point that is actually discussed at length in the introduction -- is the fact that the machine are not "random" or "sampled", but that some of them perform clever algorithms. And this is a point that the approach chosen for the paper does not (actually cannot) address. In particular, I do not feel that the paper brings us any nearer to what the authors seem to (at least implicitly) which to achieve: Implementing compressors that are based on, say, very small Turing machines that algorithmically produce an approximation of the to-be-compressed output.

4. Recommendation

Sadly, I still cannot recommend publishing this paper, even though it is an improvement over the previous version. For me, the whole approach is off the mark. However, other reviewers that lean more strongly towards a purely statistical view might see things differently.

5. Small remarks / typos

116 Raise "-n"
Algorithm 1: This algorithm is trivial in my opinion and should not be presented, just explained in one line in the text.
239 "(an approximation of Sigma)" Why is this an "approximation"? Isn't this exactly equal to |Sigma|?
241 "The NC is computed" shoudn't this rather be "P is computed"?
Algorithm 2: I made already some remarks concerning the presentation of algorithms such as this one. Especially like 2 of the algorithm is, well, intelligible only to C++-hackers. Who else will understand "mrkvTable.at(&tape[i])" ?! This is hacker speak combined with evil pointer voodoo.
Algorithm 3: Line 11 and 12: "ruleMartrix" -> "ncMatrix" (or I have utterly failed to understand what the algorithm is supposed to do)
297 "The hypothesis is"... As written earlier, I find this hypothesis nonsensical. You are, of course, entitled to verify whatever kind of hypothesis you like, but I think this is a totally wrong way of thinking about Turing machines.
317 to 319 Sorry, but why on earth do you use genomic data as "real data" in the context of a paper on Turing machines state tables? The only reason I can see is the fact that one of the authors works an a Virology Department. I do not see how this whole discussion of how variations of genomic data changes the NC values sheds any insights on, well, anything?
345, 346 Something is wrong with the phrasing of this sentence here.
366 "not universal" -> "non-universal" I guess
368 to 372: There is a lot of "interestingly" and "seems to" here. I am pretty sure that the spikes and hills arise from your numbering scheme. You write "One possible reason for this is ...". Have you checked this? Have you looked more deeply into the data here? If yes, please explain with more conviction what your findings are.
375 376 "the required bits" It is unclear to me, what these are, exactly. According to the figure, in the "inside region", around 15 bits are required "to perform compression of the tape". What does that mean? 15 bits for the whole tape seems totally wrong.
Figure 4: Sorry, I do not understand how the "normal profiles" were generated, even though I reread the description twice. What is the "compression profile" for, say, tape 32263324 (and what are these numbers, by the way? integers encoding the table? why not state the table?) at "an amplitude of 5000"?
410 The conclusion "This shows that TMs which have a higher NC exhibit a higher Martin-Löf randomness" does not follow at all form the previous sentence; indeed, I do not agree with the sentence before since I do not see that your results on synthetic data really tells us anything about Martin-Löf randomness.
521 Something is wrong with the phrasing here.
527 The last sentence of the paragraph sounds grand, but I do not see how it is supported by what has been done in the paper.

Reviewer 2 Report

This is a very interesting and well-presented paper on complexity properties of individual Turing machines and how to manipulate their output without changing their algorithmic complexity.

I think the paper is lacking information in the context of the role of initial conditions. The paper is lacking this key information because the TMs are open to arbitrary initial conditions it is a trivial result to prove without numerical research to show that a UTM is capable of unbounded statistical complexity while the UTM + a set of initial conditions remains of the same algorithmic complexity. I suspect, however, that the result is more general to individual TMs. But this should be explained and expanded significantly as it is currently not clear and it is a key aspect to understand the paper.

I also noticed an impression towards the end of the paper in the Discussion. According to the text on page 18 more measures combining algorithmic approaches and statistical measures are needed and the so-called BDM approach is given as a contrasting example. After concluding in the a paragraph in the Discussion it can be read ", which may indicate that 514 both statistical and algorithmic methods should be combined to make a better representation of data". However, BDM is itself a measure combining exactly what the authors are suggesting. I would have liked the authors to use more BDM in the core of the research as they seem to be suggesting a measure like BDM, but now that they have not done so I would suggest to characterize BDM correctly just like the authors suggest should be done. And I would stress again that I would have liked BDM more prominently used in the work given that they are advocating for measures exactly like BDM.

Reviewer 3 Report

The paper presents and interesting analysis on the relation of statistical complexity and algorithmic complexity. The methodology is well described and the results are relevant. Section 1 is a quite interesting description of the foundations and state of the art in Algorithmic Information Theory.

Some minor details:

Line 116: '2-n' => '2^{-n}'

Lines 186-187: Rewrite the sentence describing the contents of section 2 and 3. 'validation *of* our proposed method'. Possibly: 'Sections 2 and 3 include a presentation (Sect. 2) and validation (Sect. 3) of...'

Lines 334-335: 'Although less noticeable...' I cannot appreciate it in the graphics. You can provide some statistical measure about the correlation between NC and permuted blocks.

Line 365: 'The Kolmogorov complexity **of x** is defined...' Add 'of x' given that 'x' appears in line 366.

Section 4.3: I cannot find the values of #Q and #\Sigma in the comparison between Methods I and II.

Fig 5: 'NC' appears in the legend of the last graphic in the left plot, but in all other graphics appear 'Normalized Compression'. Unify the label.

Round 2

Reviewer 1 Report

This is my third review of this paper and I will not repeat everything
said in earlier reviews, but will just focus on the changes done by
the authors for the latest (r2) version and how they affect the merits
of the paper.

First, let me say that this revision is, once more, a progress on the
previous one. Some of my concerns have been addressed and the authors
clearly have invested some more time into making it clearer what their
findings are and have also actually investigated those questions I
suggested in the previous review that should be addressed.

Basically, I am still not quite convinced that "the whole approach
will lead somewhere interesting", that is, I am still not convinced
that looking at Turing machines with a few states and a binary
alphabet and then looking at the statistical properties of the outputs
they generate is overly interesting. Especially "sampling" Turing
machines with a few states and then observing how the results
behave on a statistical level seems to be missing the mark.

However, with the latest revision, I feel that the paper has now
reached a level where the authors can make a valid case that my
opinion is, well, my opinion and one can also see this in a different
way. That is, the paper now presents the results in a readable manner
and gives sensible interpretations -- it is now up to the reader to
judge whether they feel that the results are of interest to them or
not.

So, overall, I can now grudgingly recommend acceptance. Ideally,
another reviewer from the statistical community (and not, like me,
from the theoretical computer science community) should voice an
opinion on this paper. If they also think it is of interest to the
statistical community, then I would feel better about the paper's
publication. But that is of course up to the journal editors.

Small things:

199: "we show some patterns" -> "we identify some patterns"?
218: "don't" -> "do not" (at least, that's what I learned in school...)
241: ",..," -> ", ..., " This happens in several places.
241: Why is "and" it italics?
405: "computable" -> "computed"
416: ",..," -> ", ..., "
474: "an higher" -> "a higher"
540: "algorithmic" -> "algorithmically"
540: "This because" -> "This is because"?
552: "to an half" -> "to a half"
578: ",..," -> ", ..., "
606: "carnality" -> "carndinalities"?
607: "best order" -> "best-order"
639: "This algorithm has many" -> "This algorithm may have many"

Reviewer 2 Report

There is a potentially a major flaw in the reasoning of what the authors call TMs with the same algorithmic complexity. According to the authors they bound the TMs by their number of states and symbols. But this is not necessarily true and in general false. The algorithmic complexity of the TM is the shortest description of that TM which may not be the TM itself. A better approximation would be to consider only the number of rules (the number of potential rules is the number of states multiplied by the number of symbols multiplied by the halting configuration states (if any) ) used by the TM for specific input. I think this has to be addresses.

I was also confused by the Conclusion. There it says that another more 'simplistic' method, such as BDM. However, they also say that BDM can also approximate algorithmic complexity unlike their own proposal, perhaps what the authors meant was that BDM is simpler to define once all the previous steps are calculated, but I do not think it is 'simplistic'. The authors should revise this potential diminishing adjective used, which I think is not what they meant.

Also, I believe it is not true that BDM is constrained to a small number of Turing machines, it can be extended and the authors have done so too, I believe, 5 states, and not 4 like the paper suggests.

The authors claim that 'Therefore, since the purpose of this study is to quantify the statistical complexity of the TM tapes, the BDM is not suitable to be used in this case.' This seems also false even according to their own account. What BDM does is to extend the power of statistical complexity to local measurements of algorithmic complexity, so it is not that BDM is not suitable to quantify statistical complexity (it does), but that the authors are only interested in statistical complexity and therefore BDM is not suitable because goes beyond statistical complexity. I find very commendable of the authors, though, that the authors have included and contrasted their measures to a measure like BDM instead of only the more traditional and self-defeated measures such as entropy.

Notice that even more confusing when the authors claim that 'Additionally, we made a comparison with a measure that uses both algorithmic and statistical approaches (BDM) for analysis of the tapes' This cannot be true because just a few lines before they explained that BDM's main purpose was to catch some short algorithmic signals and that their approach was the one being statistical, so this is completely confusing and mixed up and has to be fixed. BDM is not meant to be only statistical, the purpose of the authors was to deviate from statistical.

The authors also contrast Levin's Universal Search and Resource-bounded algorithmic complexity to BDM but, again, BDM seems based on a function called CTM completely based on Levin's Universal Search and is clearly a Resource-bounded algorithmic complexity approach, so they should not be treating it as different. If anything BDM is probably the only numerical method put forward implementing both US and Resource-bounded. Other methods, such as Hutters and Schmidhuber's have remained on paper only because of major problems of intractability (let alone uncomputability that all these measures face).

Also notice that the same authors of the BDM came up with a similar idea to the dynamic complexity and they call it algorithmic information dynamics (http://www.scholarpedia.org/article/Algorithmic_Information_Dynamics), I think it should be said how it differs if it is only differing of the methods used (AID is quite agnostic of methods even if they also use CTM and BDM). I also found a typo 'carnality', did the authors meant 'cardinality'? If they meant canrality then I think it was never defined nor I know what does that mean.

I will be happy to support this paper once all the considerations above are taken care of, making of this a better and more precise paper, which I think it has all the potential to be. The first issue I mention is perhaps the more serious but the other ones also accumulate and are serious taken as a whole and language and method should be taken care of.

Author Response

This manuscript is a resubmission of an earlier submission. The following is a list of the peer review reports and author responses from that submission.

Round 1

Reviewer 1 Report

The paper is well written and scientifically sound, arguments are well presented and introduction is well structured with clear explanation and literature is well presented.

In my opinion the paper is worth to be published, and the only minor change that I suggest is to provide the reader with a better focus on potential applications, going over DNA sequences and discussing, by example time series analysis and image analysis.

Reviewer 2 Report

This paper aims to measure the "statistical complexity" of Turing machine "tapes" (actually, tape contents). The idea is that when even a very simple Turing machine is run (having, say, only two or three states and a binary alphabet), rather "complex" tape contents can result even when the machine is started on an initially empty tape. It thus seems like a natural question to measure how complex the contents of Turing machine tapes can become and which statistical properties these tape contents will have.

The paper starts with an extensive introduction that retraces the history of the different approaches that have been taken towards defining and measuring the amount of information and amount of randomness of symbol strings. I found this part well-written, almost like a survey that introduces readers from different fields into the topic. In the introduction, I did miss, however, a clear (or any) statement of what, exactly, are the main results and contributions of the present paper.

Following the introduction, the paper follows a methods-results-discussion structure, where the methods concern how Turing machines with a small number of states and alphabets were sampled and which parameters were chosen during their simulations. It is also explained in the methods section how the statistical complexity of bit strings is measured. (Although I am no statistician, I found these methods natural, but also quite straightforward.)

In the results section, the authors first spend some time on arguing why their proposed way of measuring complexity (using "Normalized Compression") is sensible and then present plots showing how well (or not) the tape contents produced by different Turing machines can be compressed.

The discussion section deals mostly with the question of how different compression methods might behave when applied to the tape contents; there seems to be very little interpretation of the results from the third section.

Overall, let me say that this paper is well written: Very few mistakes, extensive literature checks, sensible structure. However, I still do not see what the actual contribution of this paper actually is. It somehow seems to lie in Figure 2 and the observation that there are "clusters" of Turing machines that produce more random looking tape contents that other. Due to the naming/ordering scheme adopted by the authors these clusters seem to arise from the fact that all machines in a cluster share large parts of the transition table. Or perhaps not, who knows, I think it would have been the job of the authors to investigate this.

Perhaps my main concern lies in the fact that while the paper touches and explains a very large number of (often rather deep) approaches like Kolmogorov complexity, Martin/Loef complexity, information theory, approaches to lossless compression and so on, what has actually been done experimentally strikes me as rather simplistic. 

Sorry guys, I really do not see what the real contribution of this paper is.